# Real-Time Diffusion Policies for Games: Enhancing Consistency Policies with Q-Ensembles

**Ruoqi Zhang**[1], **Ziwei Luo**[1], **Jens Sjölund**[1], **Per Mattsson**[1], **Linus Gisslén**[2], **Alessandro Sestini**[2]

{ruoqi.zhang,ziwei.luo,jens.sjolund,per.mattsson}@it.uu.se,
{asestini,lgisslen}@ea.com

[1]**Department of Information Technology, Uppsala University, Uppsala, Sweden**
[2]**SEED - Electronic Arts (EA), Stockholm, Sweden**

## Abstract

Diffusion models have shown impressive performance in capturing complex and multi-modal action distributions for game agents, but their slow inference speed prevents practical deployment in real-time game environments. While consistency models offer a promising approach for one-step generation, they often suffer from training instability and performance degradation when applied to policy learning. In this paper, we present CPQE (Consistency Policy with Q-Ensembles), which combines consistency models with Q-ensembles to address these challenges. CPQE leverages uncertainty estimation through Q-ensembles to provide more reliable value function approximations, resulting in better training stability and improved performance compared to classic double Q-network methods. Our extensive experiments across multiple game scenarios demonstrate that CPQE achieves inference speeds of up to 60 Hz – a significant improvement over state-of-the-art diffusion policies that operate at only 20 Hz – while maintaining comparable performance to multi-step diffusion approaches. CPQE consistently outperforms state-of-the-art consistency model approaches, showing both higher rewards and enhanced training stability throughout the learning process. These results indicate that CPQE offers a practical solution for deploying diffusion-based policies in games where both multi-modal behavior modeling and rapid inference are critical requirements.

## 1 Introduction

Imitation learning and offline reinforcement learning can produce compelling and rich behaviors in complex environments. In particular, they have had early successes in domains such as robotics (Florence et al., 2022) and autonomous driving (Pomerleau, 1991) where they offer a viable alternative to hand-crafted behaviors. Similarly, the video game industry is experiencing a pressing need to improve the quality of hand-crafted behaviors, especially for large-scale environments – such as AAA games – and the need to complement human developers with automated tools to ease behavior development.

A growing body of research suggests that even simple techniques such as behavioral cloning and DAgger can enhance or even replace hand-crafted behaviors in different game domains (Bain & Sammut, 1995; Ross et al., 2011). Two common use cases are testing a game before its release and creating Non-Player Characters (NPCs) behaviors that interact with or oppose the human player (Sestini et al., 2023; Biré et al., 2024). Recent research has shown that, given a sufficiently diverse dataset of human behaviors, behavioral cloning combined with diffusion policies offers a suitable solution for recreating human-like behaviors in games (Pearce et al., 2023). This is mainly attributed to the expressiveness of diffusion model in capturing complex and multi-modal action

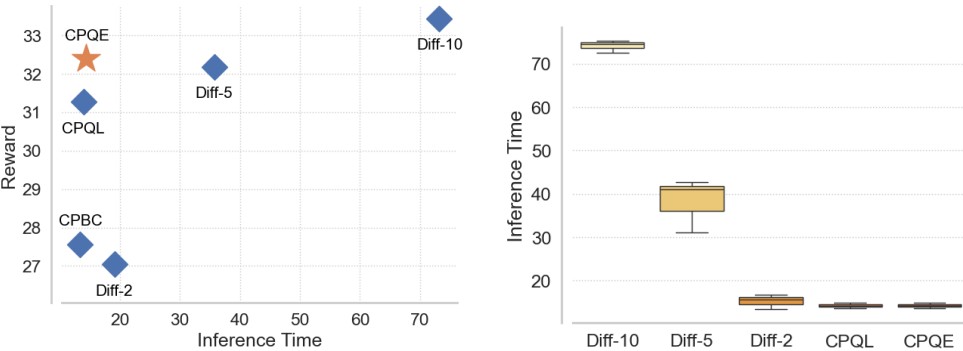

Figure 1: **Overview of the results of our work.** Left: **Inference time versus performance in Task 2** for each of the tested methods. CPQE provides a good trade-off between performance and inference time (in milliseconds). Diff-10, a diffusion policy trained with 10 denoising steps, reaches the highest reward but with the slowest inference time. CPQE outperforms the performance of Diff-5 while being 2× faster. Right: **Comparison of inference times** (in milliseconds) for each of the tested methods. CPBC is removed as it shares the same architecture and inference time as CPQL and CPQE. The figure shows the consistency-based methods as the methods with the lowest inference time.

distributions corresponding to different playstyles. For instance, a player can adopt a more proactive or more defensive playstyle in a First-Person Shooter (FPS) game. Both behaviors can lead to optimal trajectories, but they represent different styles of gameplay. To simulate the full spectrum of human players, we need a model that can synthesize multiple playstyles (Ahlberg et al., 2023).

However, diffusion models are infamously slow at inference time since they often require hundreds or thousands of denoising steps. For instance, Diffusion-X (Pearce et al., 2023) is a diffusion policy trained with a large dataset of human gameplay for the FPS game Counter-Strike: Global Offensive. Diffusion-X runs at 18 Hz compared to 200 Hz for a standard behavioral cloning approach, which precludes such approaches in game domains that usually require 60 actions per second. To reap the benefits of diffusion policies in imitation- and offline reinforcement-learning in a given game domain, it is important to run a diffusion agent sufficiently fast to produce a smooth gameplay experience while retaining the performance.

While methods like DDIM (Song et al., 2020) and DPM-Solver (Lu et al., 2022) reduce the number of sampling steps, they still require multiple iterations, making them unsuitable for real-time game applications where milliseconds matter. Consistency models (Song et al., 2023) present a way to generate action with a single step, potentially resolving the inference speed problem. Recent work (Chen et al., 2023) has advanced consistency-based policy learning with Q-learning (CPQL), though achieving robust performance across diverse game environments remains challenging. A detailed description of the related works most relevant to our study is provided in Appendix A.

As illustrated in Figure 1, traditional diffusion approaches face a clear trade-off between inference speed and performance quality. Multi-step diffusion policies achieve high reward but at the cost of significantly slower inference times, while reducing the number of inference steps – e.g. from 10 to 2 – compromises performance. This highlights the need for methods to maintain high performance while dramatically reducing inference time. To address this challenge, we propose a novel approach that combines consistency models and offline reinforcement learning with Q-ensembles (Zhang et al., 2025; An et al., 2021; Ghasemipour et al., 2022) to enable fast inference while maintaining high-quality decision-making and stable training. The consistency model can generate actions in a single step, dramatically reducing the inference time, while Q-ensembles guide policy improvement by leveraging uncertainty estimation to create lower confidence bounds on value predictions. We refer to this approach as Consistency Policy with Q-ensembles (CPQE).

## 2  Preliminaries

In this section, we first present essential background on *Imitation Learning* and *offline Reinforcement Learning* (RL) and then introduce two classes of generative models: *Diffusion Models* and *Consistency Models*. These preliminaries lay the foundation for our methodology in Section 3.

### 2.1  Imitation- and Offline Reinforcement-Learning

We consider a Markov decision process defined by a state space $\mathcal{S}$, action space $\mathcal{A}$, transition probabilities $P(s' \mid s, a)$, reward function $r(\cdot, \cdot)$, and discount factor $\gamma \in [0, 1)$. In standard reinforcement learning, an agent interacts with this environment to learn an optimal policy. However, in offline RL, the agent must learn solely from a static dataset, $\mathcal{D}_{\text{offline}} = \{(s, a, r, s')\}^N$, where each of the $N$ tuples contains a state $s$, action $a$, reward $r$, and next state $s'$, collected by some unknown *behavior policy*. Since no further environment interactions are allowed, a common approach is to imitate expert data. In imitation learning, an agent learns by replicating the action of the expert given the states – the reward is not needed. In contrast, *offline* or *batch* RL aims to learn a value function that tries to maximize the expected cumulative discounted reward. While offline RL agents often exhibit better generalization, both approaches share the goal of generating optimal behavior. In real-world settings, datasets are often multimodal; even when collected from the same expert, behaviors may vary significantly in similar states. Thus, we require a highly expressive policy to capture this variability.

### 2.2  Diffusion Models and Consistency Models

*Diffusion models* (Ho et al., 2020; Song et al., 2021) are a class of generative models that produce samples by *reversing* a noising process. In the forward noising process, data $x_0 \sim p_{\text{data}}(x)$ is progressively corrupted with Gaussian noise. This can be expressed as a stochastic differential equation:

$$dx_k = \mu(x_k, k)dk + \sigma(k)dw_k, \tag{1}$$

where $k \in [0, K]$, $K$ is a fixed positive constant, $\{w_k\}_{k \in [0, K]}$ is a standard Brownian motion, and $\mu$ and $\sigma$ are drift and diffusion coefficients respectively. Then, starting from $x_K$, a sample from the data distribution that approximates $x_0$ can be generated by solving the reverse process with the probability flow ODE:

$$dx_k = \left[ \mu(x_k, k) - \frac{1}{2}\sigma(k)^2 \nabla \log p_k(x_k) \right] dk, \tag{2}$$

where $\nabla \log p_k(x_k)$ is the score function of $p_k(x)$, which is unknown but can be approximated by a neural network via score matching (Ho et al., 2020; Song et al., 2021). In this paper, we follow the work of Karras et al. (Karras et al., 2022) and let $\mu(\cdot) = 0$ and $\sigma(k) = \sqrt{2k}$. Then, with a trained score model $s_\theta(x, k) \approx \log p_k(x_k)$, an empirical estimate of the probability flow ODE takes the form:

$$dx_k = -ks_\theta(x, k)dk. \tag{3}$$

The sampling can be done by first sampling $\hat{x}_K = \mathcal{N}(0, K^2 I)$ as the initial values of the empirical probability flow ODE and then using any numerical ODE solver – for example, the Euler (Song et al., 2020; 2021) and Heun solvers (Karras et al., 2022) – to obtain the solution $\hat{x}_0$. Such approaches can capture complex, multi-modal distributions if $s_\theta$ is general enough. In an RL context, we can let $x \equiv a$ denote actions, so that diffusion models represent the action distribution under each state. The resulting learned generative model can then sample actions consistent with offline data. However, the reverse process can be slow, as it requires several denoising steps.

*Consistency models* (Song et al., 2023) offer an alternative formulation aiming to reduce or eliminate the multi-step reverse pass. Instead of iterating over many small increments in Equation 3, a consistency model $f_\theta(x_k, k)$ directly infers $x_0$ from $x_k$ in **one** (or very few) steps: $x_0 \approx f_\theta(x_k, k)$.

The training enforces a *consistency condition*, namely that the model produces the same clean output regardless of which $k$ is used as input. By penalizing deviations from this condition, the model learns to bypass iterative sampling. Consistency models are appealing in RL domains where inference speed is crucial, as in games. However, they require careful design to ensure adequate coverage of the data distribution without multi-step denoising.

## 3    Consistency Policies for Offline RL

Consistency policies mitigate the problem posed by the multi-step denoising process in diffusion models by providing single-step inference, substantially improving inference speed. While consistency models offer efficient single-step inference, they are typically less expressive than diffusion models when trained via pure imitation learning. This expressiveness gap can lead to suboptimal policy performance in complex environments. However, we demonstrate that incorporating Q-learning and ensembles into the consistency policy training process substantially improves performance by both providing additional learning signals beyond behavioral cloning and mitigating the inherent uncertainty in offline RL.

### 3.1    Consistency Policies for Behavior Cloning

Given the advantages of single-step inference described above, we now detail how to implement consistency policies using behavior cloning. Following the work by Song et al. (Song et al., 2023), a consistency policy $f_\theta(a_k, k \mid s)$ is a parameterized neural network trained to satisfy the consistency condition that maps any noisy action $a_k$ at noise level $k$ back to the same clean action $a$:

$$\pi_\theta(a \mid s) = f_\theta(a_k, k \mid s)$$
$$= c_{\text{skip}}(k)\, a_k + c_{\text{out}}(k)\, F_\theta(a_k, k \mid s), \tag{4}$$

where $F_\theta$ is a neural network predicting the denoised action. The terms $c_{\text{skip}}(k)$ and $c_{\text{out}}(k)$ are predetermined differentiable functions that enforce the boundary condition of these models. We use the same functions for $c_{\text{skip}}$ and $c_{\text{out}}$ as defined in (Song et al., 2023). Specifically, these functions are constructed to ensure $c_{\text{skip}}(\epsilon) = 1$ and $c_{\text{out}}(\epsilon) = 0$ when the noise level $k$ reaches a small positive constant $\epsilon$. This maintains differentiability at $k = \epsilon$ when $F_\theta$ is differentiable. We terminate the reverse process at $k = \epsilon$ rather than $k = 0$ to prevent numerical instabilities.

**Training:** There are two main approaches to train consistency models. One is consistency distillation which requires a pretrained diffusion model and another one is consistency training which trains from the scratch. In our work, we focus on the latter approach. In practice, the probability flow ODE is discretized into $N - 1$ sub-intervals with boundaries $k_1 = \epsilon < k_2 < \cdots < k_N = K$. The values can be determined as $k_n = (\epsilon^{1/\rho} + \frac{n-1}{N-1}(K^{1/\rho} - \epsilon^{1/\rho}))^\rho$ with $\rho = 7$ (Karras et al., 2022). For consistency training, we leverage an unbiased estimator (Song et al., 2023):

$$\nabla \log p_{k_n}(a_{k_n}) = -\mathbb{E}\left[\frac{a_{k_n} - a_0}{k_n^2} \mid a_{k_n}\right], \tag{5}$$

where $a_0 \sim \mathcal{D}_{\text{offline}}$, $n \sim \mathcal{U}[1, N-1]$ and $a_{k_n} \sim \mathcal{N}(a_0; k_n^2 I)$. Thus, given $a_0$ and $a_{k_n}$, the score $\nabla \log p_k(a_{k_n})$ can be estimated with $\frac{a_{k_n} - a_0}{k_n^2}$. This eliminates the need for a pretrained score model and allows us to define the consistency training loss:

$$\mathcal{L}_{\text{CT}}(\theta) = \mathbb{E}\left[\lambda(k_n)\left\|f_\theta(a_{k_n}, k_n \mid s) - f_{\bar{\theta}}(\hat{a}_{k_{n+1}}, k_{n+1} \mid s)\right\|_2^2\right], \tag{6}$$

where the expectation is taken w.r.t $(s, a_0) \sim \mathcal{D}_{\text{offline}}$, $\bar{\theta}$ is the exponential moving average over the past values of $\theta$, and $\lambda(\cdot) > 0$ is a positive weighting function. We found that, in practice, Equation 6 can lead to training instability and thus added a reconstruction loss:

$$\mathcal{L}_{\text{RC}}(\theta) = \mathbb{E}\left[\left\|f_\theta(a_{k_n}, k_n \mid s) - a_0\right\|_2^2\right]. \tag{7}$$

The reconstruction loss explicitly drives the consistency model to recover the original clean action rather than indirectly achieving recovery through consistency conditions. Finally, our consistency loss is $\mathcal{L}_{\text{consistency}}(\theta) = \mathcal{L}_{\text{RC}}(\theta) + \mathcal{L}_{\text{CT}}(\theta)$.

**Single-Step Inference:** Once trained, the consistency policy $f_\theta$ can sample actions in one forward pass: $a \mid s = a_\epsilon = f_\theta(a_K, K \mid s)$, where $a_K \sim \mathcal{N}(0, K^2 I)$.

### 3.2 Consistency Policy with Q-learning

While the consistency training approach described above provides efficient inference, it remains limited by the behavioral cloning paradigm that only imitates actions from the dataset without leveraging reward information. CPQL (Chen et al., 2023) integrates a single-step consistency policy with a conservative Q-learning framework to enhance offline RL. In CPQL, a Q-network $Q_\phi(s, a)$ is trained in a classical way with the Bellman operator and the double Q-learning trick:

$$\mathcal{L}_Q(\phi) = \mathbb{E} \left\| r + \gamma \min_{i=1,2} Q_{\bar{\phi}_i}(s', a'_\epsilon) - Q_\phi(s, a) \right\|^2, \tag{8}$$

where $\phi$ is the parameters of the Q-network, the expectation is taken over $(s, a, r, s') \sim \mathcal{D}_{\text{offline}}$ and $a'_\epsilon \sim \pi_\theta(s')$, and $Q_{\bar{\phi}_i}$ are target networks. CPQL adds the learned Q-function to the policy loss as regularizer. The overall policy loss then becomes:

$$\mathcal{L}_\pi(\theta) = \mathcal{L}_{\text{consistency}}(\theta) - \alpha \mathbb{E} \left[ Q_\phi\big(s, f_\theta(a_k, k \mid s)\big) \right], \tag{9}$$

where $\alpha = \eta / \mathbb{E}_{(s,a) \sim \mathcal{D}_{\text{offline}}}[Q(s, a)]$ balances behavior cloning against policy improvement.

### 3.3 Consistency Policy with Q-ensembles (CPQE)

In practice, Q-function estimates in offline setting suffer from high variance and systematic overestimation, particularly for out-of-distribution actions due to the distributional shift between offline data and policy-generated actions. To mitigate these estimation errors, we propose CPQE, which use Q-ensembles (An et al., 2021; Zhang et al., 2025) to better capture uncertainty in value estimates. The Q-ensemble consists of $M$ Q-networks $Q_{\phi_1}, \ldots, Q_{\phi_M}$ trained with different initialization and independent targets to maximize the diversity of the ensemble (Ghasemipour et al., 2022). More specifically, the Q-ensemble loss for $m$-th Q-network is defined as:

$$\mathcal{L}_{QE}(\phi_m) = \mathbb{E} \left[ Q_{\phi_m}(s, a) - y_m(r, s' \mid \pi_\theta) \right]$$
$$y_m = r + \gamma Q_{\bar{\phi}_m}(s', \pi_\theta(a_k, k \mid s)), \tag{10}$$

where the expectation is taken over $(s, a, r, s') \sim \mathcal{D}_{\text{offline}}$, $Q_{\phi_m}$ and $Q_{\bar{\phi}_m}$ are the Q-network and target network for the $m$-th Q-network, respectively. The final Q-ensemble loss is the average of the loss of each Q-network. Then, we use the ensemble mean or the Lower Confidence Bound (LCB) of the Q-values to update the policy network. The pessimistic LCB of the Q-values is defined as:

$$Q_\phi^{\text{LCB}}(s, a) = \mathbb{E}[Q_{\phi_m}(s, a)] - \beta \sqrt{\mathbb{V}[Q_{\phi_m}(s, a)]}, \tag{11}$$

where the expectation and the variance are w.r.t the ensemble and $\beta > 0$ is a hyperparameter that controls the pessimism level. Finally, we update the Q-values estimate in Equation 9 with $Q_\phi^{\text{LCB}}(\cdot, \cdot)$. The full algorithm is outlined in Appendix B.

## 4 Experiments and Results

In here, we first present the environment and tasks used to evaluate our method. Second, we describe the baselines used to compare our approach. Finally, we present the main results of our study, which show that CPQE has the lowest inference time while achieving the same performance as the baselines.

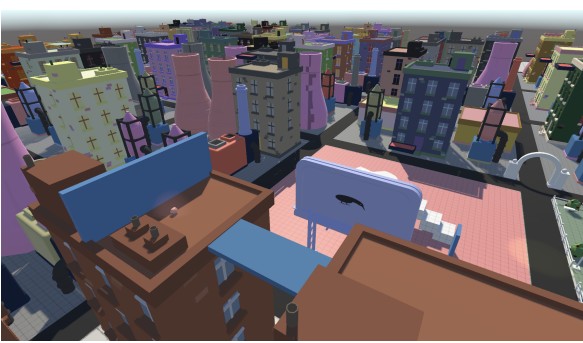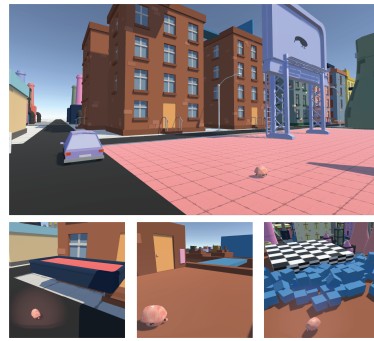

Figure 2: **Overview of the environment used in this study**. Left: The environment represents a 3D open- and procedurally generated-world. More details are provided in Appendix C. Right: **Example of a trajectory in *Task 2***. The agent's starting position is on the ground, and it has to navigate to a elevator, wait for it to come down and jump over it. Once it is up on the building, the agent needs to cross a bridge: if it falls, there is no way to get back on track. The agent has to shoot at a destructible wall in order to reveal the goal location. This example is showing the CPQE policy acting in the environment. A close-up figure is available in Appendix C.

## 4.1 Experimental Setup

**Environment and tasks.** Figure 2 shows the environment used in this study. It is an open-world city simulation originally proposed by Sestini et al. (Sestini et al., 2023). More details about the environment, such as action- and state-space, are provided in Appendix C. The environment is created using the Unity 3D game engine (Juliani et al., 2018). We implement the inference code in C# to run inference and collect sampling time directly within the engine. The policies are trained in Python using the PyTorch library. After training, we export the neural networks using the ONNX format. We use the Sentis package in Unity to load and run the model in game.

Within the environment, one can define different tasks. In this work, we use two variations: *Task 1*, which requires the agent to reach a goal position on top of a building relatively close to the agent's starting position. This is a simple task where the primary metric is inference time rather than performance; and *Task 2*, which is a relatively complex task. The agent has multiple intermediate goals: use an elevator, cross over a bridge, destroy a wall by shooting at it, and arrive at the goal location. This task is the same as *Use Case 2* proposed by Sestini et al. (Sestini et al., 2023), and it is described in Figure 2. In this task, we focus on the ratio between performance and inference time.

**Baselines.** We evaluate CPQE against the following baselines: (1) **_Diff-t_**: a diffusion model trained with $t \in \{10, 5, 2\}$ denoising steps. This is our main baseline, with *Diff-10* achieving the highest performance but with a longer sampling time. Ideally, we aim for a method that achieves the same performance but with a lower inference time; (2) **_CPBC_**: a consistency policy trained with a behavioral cloning objective; and (3) **_CPQL_**: a consistency policy trained with offline RL and Q-learning; For each task, we first train a policy using each method. As we will see in the next section, our approach and all the baselines use a U-Net-based policy. After training, we run the policy in the game engine and collect the cumulative episodic reward and average inference time over 10,000 environment steps. We repeat each experiment for 3 different seeds.

**Experiments Details.** For our training data, we collected 200,000 state-action transitions for each task from a soft actor-critic (Haarnoja et al., 2018) agent that was trained until convergence. We gathered the data by running evaluation episodes in the environment with the trained agent. Following the work by Chi et al. (Chi et al., 2023), we use the closed-loop action-sequence prediction to promote consistent action generation and enhance execution robustness in both diffusion and consistency policies. More specifically, at each time step $t$, the policy takes the latest 2 observations as state and predicts $N_p$ future actions while only $N_a$ steps of actions are executed in the game environment. For

| | Task 1 | | Task 2 | |
|---|---|---|---|---|
| **Method** | **Reward ↑** | **Inference Time ↓** | **Reward ↑** | **Inference Time ↓** |
| Diff-10 | $4.58 \pm 2.57$ | $74.08 \pm 1.13$ | $33.44 \pm 0.52$ | $73.16 \pm 1.12$ |
| Diff-5 | $6.31 \pm 1.49$ | $38.08 \pm 5.11$ | $32.18 \pm 0.41$ | $35.78 \pm 3.10$ |
| Diff-2 | $2.99 \pm 3.63$ | $15.29 \pm 1.40$ | $27.04 \pm 0.46$ | $16.08 \pm 1.12$ |
| CPBC | $5.33 \pm 0.68$ | $14.13 \pm 0.52$ | $27.56 \pm 1.48$ | $14.31 \pm 0.61$ |
| CPQL | $5.79 \pm 0.56$ | $14.20 \pm 0.48$ | $31.27 \pm 1.59$ | $13.97 \pm 0.40$ |
| CPQE | $5.80 \pm 0.49$ | $14.34 \pm 0.45$ | $32.39 \pm 1.15$ | $14.32 \pm 0.43$ |

Table 1: **Performance and inference time for both tasks** are presented for each of the tested methods. Higher reward values are better, while lower inference times are preferable. In Task 1, which is a simple task, all the methods except Diff-2 reach a very high reward. However, CPQE has a $5\times$ faster inference time than Diff-10. In Task 2, although CPQE does not achieve the performance level of Diff-10, it outperforms Diff-5 but with $2\times$ faster inference. In both tasks, CPBC does not reach the performance of CPQE, even with the same inference time.

task 1, we set $N_p = 4$ and $N_a = 4$. For the more complex task 2, we set $N_p = 16$ and $N_a = 8$. Our primary model employs a 1D U-Net (Ronneberger et al., 2015) that effectively captures the spatial information from the 3D semantic map through conditional residual blocks with Feature-wise Linear Modulation (FiLM) conditioning (Perez et al., 2018). Additionally, we run an ablation study replacing the U-Net based policies of CPQL and CPQE with a MLP. We refer to these ablations as CPQL-MLP and CPQE-MLP. For these, we substitute this with a three-layer MLP backbone with 512 units per hidden layer while maintaining identical time embedding. More details about the architectures and the hyper-parameters used in this study are provided in Appendix D.

### 4.2   Results

In this section we will describe the results for each of the task defined in Section 4.1. Table 1 summarizes the results.

*Task 1.*   According to Table 1, this task proves to be relative straightforward, with almost all policies achieving a high performance, except *Diff-2*. For this task, we primarily focus on comparing computational efficiency across different methods. Our findings indicate that consistency-based approaches, CPBC, CPQL, and CPQE execute substantially faster than diffusion-based methods. This efficiency advantage is mainly due to the consistency models requiring just a single inference step, in contrast to multiple steps required by diffusion approaches, e.g. 10 steps used by *Diff-10*. The three consistency-based methods demonstrate identical inference times since they employ the same underlying network architecture and all execute just one forward pass. The results show that our method produces a policy that runs in-game at 60 Hz, meaning on an average of 60 Frames Per Second (FPS). While this is not yet sufficient for deploying diffusion policies in games without compromises, as other systems also require computational time within each frame, it represents a significantly faster inference time compared to the 20 Hz of the state-of-the-art method *Diffusion-X* (Pearce et al., 2023). We believe our CPQE approach is a significant step towards running diffusion policies in games. However, it is essential to ensure that the faster inference time does not adversely affect the agent's performance. Therefore, we conducted the same experiments with a more complex task.

*Task 2.*   As Table 1 shows, reducing inference time does not significantly impact performance even in a complex task. Our CPQE approach achieves better performance than the *Diff-5* baseline and comparable performance to the *Diff-10* baseline, while maintaining a faster inference time. In particular, our approach can run 2 times faster than *Diff-5*, achieving better performance, and 3 times faster than *Diff-10*, achieving similar results. This demonstrates that our method can provide both fast and effective diffusion policies. CPQE outperforms CPQL, demonstrating that our approach of leveraging multiple Q-networks to provide more reliable value estimation improves performance over the state-of-the-art. Interestingly, the CPBC approach has a performance similar to *Diff-2*. Although

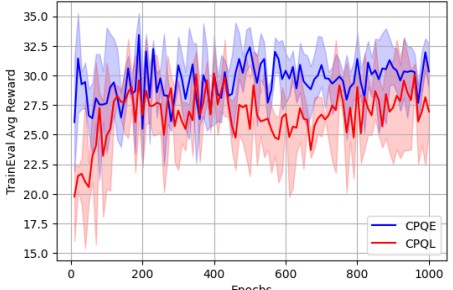

Figure 3: **Training performance of CPQL versus CPQE** on Task 2s. CPQE demonstrates both higher returns and enhanced training stability throughout the training.

| Method | Task 1 | Task 2 |
|---|---|---|
| CPQL | $5.79 \pm 0.56$ | $31.27 \pm 1.59$ |
| CPQL-MLP | $1.96 \pm 0.30$ | $17.08 \pm 1.91$ |
| CPQE | $5.80 \pm 0.49$ | $32.39 \pm 1.15$ |
| CPQE-MLP | $1.61 \pm 0.12$ | $19.77 \pm 3.02$ |

Table 2: **Performance comparison between U-net and MLP**. The ablated versions of CPQE and CPQL are unable to reach the same performance as the U-Net based policies.

CPBC, CPQL, and CPQE share the same underlying architecture, both CPQL and CPQE leverage Q-learning techniques to boost the policy learning via offline RL. Our experiments indicate that Q-learning is crucial for a consistent policy to match the performance of a diffusion policy with multiple denoising steps. Figure 1 compares inference times for each of the tested methods and summarizes the performance of all agents, highlighting that our method offers the best trade-off. Figures 2 and Figure 4 illustrate an example of a CPQE trained agent solving *Task 2*.

***Comparing CPQE to CPQL.*** Figure 3 demonstrates the significant advantages of our Q-ensemble approach over standard Q-learning when combined with consistency models. As shown in the plot, CPQE consistently outperforms CPQL across the entire 1,000 epoch training period on Task 2. More importantly, while both methods exhibit variance during early training stages, CPQE demonstrates superior stability in later epochs, with less fluctuations in performance. This stability is particularly crucial for game environments where consistent agent behavior is essential for player experience. The enhanced performance is due to the ability of Q-ensembles of providing more accurate value function estimates through uncertainty quantification, which improves policy optimization. These results suggest that replacing double Q-networks with ensembles helps address training stability challenges that hinder consistency model training in offline RL settings. Note that the the results in Figure 3 show the mean performance at each training iteration, while the results shown in Table 1 represent the mean of *best* evaluation which are slightly higher than the value on the plot.

***Comparing U-Net to MLP.*** Table 2 highlights the benefits of using a U-Net-based policy network for training. We train an ablated version of CPQE and CPQL, referred to as CPQE-MLP and CPQL-MLP, where the U-Net was replaced with a multi-layer perceptron (details in Section 4.1), while keeping the training process identical. Although the MLP versions run faster due to their simpler neural network structure – 7 milliseconds of the MLP variants compared to 14 milliseconds of the U-Net based ones – they underperforms U-Net based policies in both tasks. This indicates that a more sophisticated network architecture strikes a good balance between performance and inference time.

## 5 Conclusion

In this paper, we present CPQE, Consistency Policy with Q-Ensembles, which combines consistency models and Q-ensembles to achieve fast inference, stable training, and enhanced performance in game environments. Our experiments demonstrate that CPQE achieves comparable performance to multi-step diffusion policies while significantly reducing inference time – operating at 60 Hz compared to the 20 Hz of state-of-the-art methods such as Diffusion-X. This represents a substantial step toward making diffusion models practical for real-time game applications. Thanks to Q-ensembles, CPQE provides more reliable value function estimates through uncertainty quantification. This approach not only improves training stability but also enhances performance.

**Limitations and Future Work.** Despite the promising results, CPQE still faces limitations. While our approach significantly reduces inference time, reaching 60 Hz may still be insufficient for some high-performance games. Additionally, our implementation requires more computation resources than traditional approaches during training due to the ensemble of Q-networks and the consistency model. We would like to explore more efficient methods to improve the stability of consistency model training and applications to multi-agent scenarios for creating more realistic and coordinated NPC behaviors in complex game environments.

## A    Related Work

The challenge of creating agents with diffusion models, especially in games, has gained recent interest. However, these models are infamously known for being slow at inference time.

### A.1    Generative Models in Games

Video games have often been used as a testbed for decision-making agents (Wurman et al., 2022; Berner et al., 2019). Recent work has shown how imitation learning and offline reinforcement learning can help develop agents not only for playing video games but also for being part of their design. Sestini et al. proposed an approach based on DAgger to let designers create testing agents (Sestini et al., 2023). This approach was later improved by Biré et al., who used random network distillation to query the designer more efficiently (Biré et al., 2024). Pearce et al. used imitation learning to train an agent that can play the game Counter-Strike: Global Offensive (Pearce & Zhu, 2022). The agent was subsequently improved by Diffusion-X, a diffusion model trained with the same dataset to achieve human-level performance in the game (Pearce et al., 2023). Although diffusion models have been mainly used for generating images and, in some cases, even entire games (Decart et al., 2024; Valevski et al., 2024), a growing body of literature has shown that diffusion models can be used for training decision-making agents, especially when policy diversity is a requirement (Pearce et al., 2023; Høeg et al., 2024).

### A.2    Accelerating Inference in diffusion models

Despite their impressive generative capabilities, diffusion models require substantial computational resources during inference. Three main approaches have been developed to mitigate this limitation. First, sampling efficiency techniques like DDIM (Song et al., 2020) and DPM-Solver (Lu et al., 2022) mathematically reformulate the diffusion process to reduce required sampling steps, yielding up to 10x speedup. Second, knowledge distillation methods such as Progressive Distillation (Salimans & Ho, 2022) train smaller student models to replicate teacher outputs with fewer steps, achieving comparable quality with as few as 4 iterations. Third, dimensionality reduction approaches including Latent Diffusion Models (Rombach et al., 2022; Podell et al., 2023) operate in compressed latent spaces rather than pixel space, using VAE-based compression to significantly lower computational demands while maintaining generation quality, though the number of diffusion steps remains unchanged. In this paper, we employ the Consistency Model (Song et al., 2023) as our policy to achieve one-step diffusion inference. To the best of our knowledge, CPQL (Chen et al., 2023) is most closely related to our work, as it combines Q-learning with a consistency model policy. However, we found that CPQL exhibits training instability and therefore we propose CPQE, which uses Q-ensembles to learn more accurate Q-value functions. Our approach leverages the LCB of estimated Q-values to train our policy, resulting in greater stability and better performance.

## B    Algorithm

To mitigate the problem of high variance and systematic overestimation, particularly for out-of-distribution actions, CPQE uses !-ensembles to better capture value estimates. The Q-ensembles

---

**Algorithm 1** CPQE for Game

---

Initialize the policy network $\pi_\theta$, the Q-ensemble $\{Q_{\phi_m}\}_{m=1}^M$, and the target networks $\pi_{\bar\theta}$, $\{Q_{\bar\phi_m}\}_{m=1}^M$

**for** each iteration **do**
    Sample a batch of data $\mathcal{D}_B = \{s, a, r, s'\} \sim \mathcal{D}_{\text{offline}}$
    **# Ensemble-Q learning**
    Update the Q-networks by Equation 10
    **# Consistency Policy Learning**
    Sample $a_K \sim \mathcal{N}(0, K^2 I)$ and then get the action to take by
    Equation 4
    Update $\pi_\theta$ by minimizing Equation 9 with $Q_\phi^{\text{LCB}}(s, a_\epsilon)$ in Equation 11
    **#Update the target networks**
        $Q_{\bar\phi_i} \leftarrow \tau Q_{\bar\phi_m} + (1-\tau)Q_{\phi_m}$ for $m \in \{1, \dots, M\}$
        $\pi_{\bar\theta} \leftarrow \tau\pi_{\bar\theta} + (1-\tau)\pi_\theta$
**end for**

---

consists of $M$ Q-networks $Q_{\phi_1}, \dots, Q_{\phi_M}$ trained with different initialization and independent targets to maximize the diversity of the ensemble. The full algorithm is outlined in Algorithm 1.

## C    Environment Details

In this section, we provide more details on the environment showed in Figure 2 It is an open-world city simulation originally proposed by Sestini et al. (Sestini et al., 2023). In this environment, the agent has a continuous action space of size 5, consisting of: two actions representing the movement vector, one action for strafing left and right, one action for shooting and one action for jumping. Each action is normalized between $[-1, 1]$, while the latter two are discretized in the game. The state space consists of a game goal position, represented as the $\mathbb{R}^2$ projections of the agent-to-goal vector onto the $XY$ and $XZ$ planes, normalized by the gameplay area size, along with game-specific state observations such as the agent's climbing status, contact with the ground, presence in an elevator, jump cooldown, and weapon magazine status. Observations include a list of entities and game objects that the agent should be aware of, e.g., intermediate goals, dynamic objects, enemies, and other assets that could be useful for achieving the final goal. For these entities, the same relative information from agent-to-goal is referenced, except as agent-to-entity. Additionally, a 3D semantic map is used for local perception. This map is a categorical discretization of the space and elements around the agent. Each voxel in the map carries a semantic integer value describing the type of object at the corresponding game world position. For this work, we use a semantic map of size $5 \times 5 \times 5$. In this environment, an episode consists of a maximum of 1000 steps. An episode is marked a success if the agent reaches the goal before the timeout. The environment is particularly meaningful for this study because it uses standard state- and action-spaces for developing RL agents for in-game NPCs. In fact, it is common to avoid using image-based agents, as they are too expensive at runtime, and instead use floating-point information gathered from the game engine as the state space, similar to traditional scripted game-AI. While there are no specific preferences between discrete or continuous action-spaces, it is common to use the latter in this domain (Wurman et al., 2022; Gillberg et al., 2023). Figure 4 shows a CPQE-based agent solving Task 2.

## D    Hyper-parameters

As previously mentioned, our primary model employs a 1D U-Net (Ronneberger et al., 2015) that effectively captures the spatial information from the 3D semantic map through conditional residual blocks with Feature-wise Linear Modulation (FiLM) conditioning (Perez et al., 2018). The U-Net architecture consists of a downsampling block with channel dimensions [32, 64], a bottleneck, and

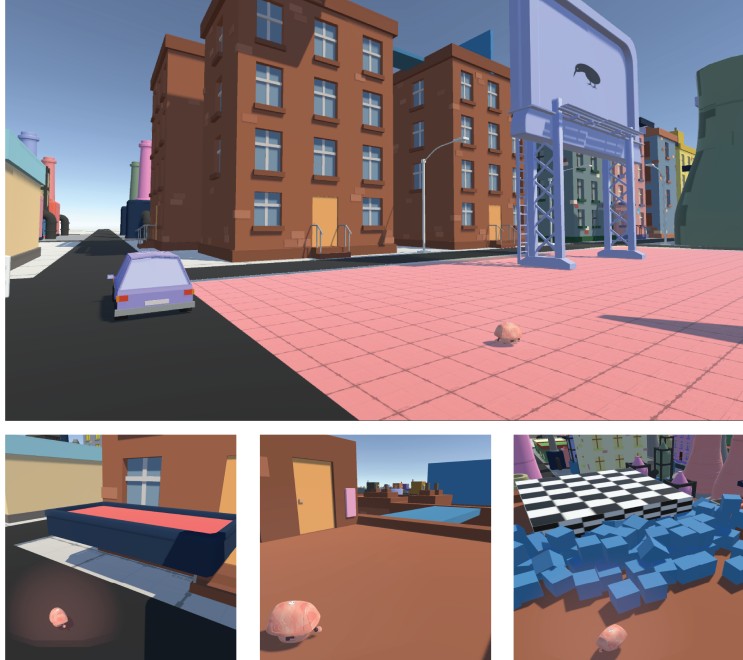

Figure 4: **Example of a trajectory in *Task 2***. The agent's starting position is on the ground, and it has to navigate to a elevator, wait for it to come down and jump over it. Once it is up on the building, the agent needs to cross a bridge between two buildings: if it falls, there is no way to get back on track. The agent has to shoot at a destructible wall in order to reveal the goal location. This example is showing the CPQE policy acting in the environment.

an upsampling block with skip connections. Each level includes conditional residual blocks that incorporate state observations and diffusion step embeddings.

Additionally, we run an ablation study replacing the U-Net based policies of CPQL and CPQE with a MLP. We refer to these ablations as CPQL-MLP and CPQE-MLP. For these, we substitute this with a three-layer MLP backbone with 512 units per hidden layer while maintaining identical time embedding. Both the policy and the value network are trained using the Adam optimizer with a learning rate of $10^{-4}$. The training epoch is set to 250 for task 1 and 500 for task 2. For our CPQE methods, we train an ensemble of 16 Q-networks with different random initializations. The value of $\eta$ is set to $1.0$ for CPQL and $0.5$ for our method CPQE. The LCB coefficient $\beta$ is set to $1.0$ for both tasks, balancing conservatism with performance. All models are trained on an NVIDIA A100 GPU with 40GB memory, while inference times are recorded using a Macbook Air with M2 processor and 16GB of RAM.

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
