# OpenReview forum: "Real-Time Diffusion Policies for Games: Enhancing Consistency Policies with Q-Ensembles"
_rl-conference.cc/RLC/2025/Workshop/RLVG — RLVG Workshop - RLC 2025_

### Official Review · Reviewer_cbuS · 2025-06-14
**Good Contribution**

**Rating:** 4
**Confidence:** 3

**Summary:**

The paper introduces CPQE (Consistency Policy with Q-Ensembles), a method that combines consistency models with Q-ensembles to enable fast and stable policy learning for real-time games. CPQE achieves single-step action generation with significantly faster inference (up to 60 Hz) than traditional diffusion models, while maintaining strong performance. By leveraging Q-ensembles for uncertainty-aware value estimation, CPQE improves training stability and outperforms prior consistency-based approaches, offering a practical solution for deploying diffusion-like policies in interactive game environments.

**Strengths:**

- The paper is well written and easy to follow.
- The paper proposes a practical contribution to real-time decision-making in games, combining the expressiveness of diffusion models with the speed of consistency models.
- CPQE's use of Q-ensembles provides more stable and accurate value estimates, leading to improved performance and training reliability.
- The approach is well-validated through experiments in complex, realistic game environments, showing clear advantages over state-of-the-art baselines in both inference speed and reward.
- The method is scalable and general enough for broader applications in offline RL and imitation learning.

**Weaknesses:**

- While the choice of consistency models helps achieve fast, single-step inference, the paper does not justify well enough why alternatives like shortcut models [1] were not explored.
- The method resembles Diffusion Q-learning [2] (DQL), but enhanced with an ensemble (and less diffusions steps). I believe that the authors should cite this work.
- It would be valuable to integrate a DQL-ensemble version as an "oracle" baseline to set an upper bound on the agent's performance. The comparison with standard diffusion seems not entirely fair, as CPQE benefits from an additional RL loss, which diffusion baselines do not leverage.
- The performance of the SAC agent used to collect the data is missing, which prevents the reader from assessing whether the proposed method surpasses the demonstration policy or merely matches it.
- The evaluation is limited to only two tasks. Additional tasks or environments would help demonstrate the method’s robustness and generalizability.

[1] Frans, Kevin, et al. "One step diffusion via shortcut models." arXiv preprint arXiv:2410.12557 (2024).

[2] Wang, Z., et al. "Diffusion policies as an expressive policy class for offline reinforcement learning." ICLR 2023

**Best Paper Nomination:**

No

**Claims:**

Yes, the authors provide enough evidence to support their claims.

**Suggestions:**

Overall, this paper makes a meaningful and timely contribution to advancing diffusion-inspired policies for real-time games. With just a few clarifications, additional baselines, and a more extensive evaluation across diverse tasks, the work could be even stronger and have broader impact.

---

### Official Review · Reviewer_icSU · 2025-06-15
**Review of Real-Time Diffusion Policies for Games: Enhancing Consistency Policies with Q-Ensemble**

**Rating:** 2
**Confidence:** 3

**Summary:**

Faster diffusion-based policies capable of modelling multimodal Q-values with better performance.

**Strengths:**

Clear exposition of the method

**Weaknesses:**

CPQE is a combination of existing methods but does not bring novel insights and does not outperform existing methods. More standard environments with stronger baselines would also be a good addition.

**Best Paper Nomination:**

No

**Claims:**

No, results do not really suggest that CPQE is significantly better than CPQL. Given the incremental nature of CPQE over CPQL, I would expect stronger results to validate the approach.

**Suggestions:**

See weaknesses

---

### Decision · Program_Chairs · 2025-06-19

**Decision:**

Accept

**Comment:**

The paper proposes CPQE (Consistency Policy with Q-Ensembles), which combines fast one-step consistency models with Q-ensembles to improve uncertainty estimation, training stability, and performance in game agents.
This work provides contributions in an important area, specifically real-time decision-making in games, and the claims are supported by empirical evidence on complex game environments.
The reviews recommend including and discussing additional relevant work, as well as providing a thorough evaluation of the SAC agent.
We strongly encourage the authors to address these points in the camera-ready version.